# Excess burden of respiratory and abdominal conditions following COVID-19 infections during the ancestral and Delta variant periods in the United States: An EHR-based cohort study from the RECOVER program

**Jay K. Varma**[1]*, **Chengxi Zang**[1], **Thomas W. Carton**[3], **Jason P. Block**[4], **Dhruv J. Khullar**[1,2], **Yongkang Zhang**[1], **Mark G. Weiner**[1], **Russell L. Rothman**[5], **Edward J. Schenck**[2], **Zhenxing Xu**[1], **Kristin Lyman**[1], **Jiang Bian**[6], **Jie Xu**[6], **Elizabeth A. Shenkman**[6], **Christine Maughan**[7], **Leah Castro-Baucom**[8], **Lisa O'Brien**[7], **Fei Wang**[1], **Rainu Kaushal**[1], on behalf of the RECOVER Consortium[¶]

1 Department of Population Health Sciences, Weill Cornell Medicine, New York, New York, United States of America, 2 Department of Medicine, Weill Cornell Medicine, New York, New York, United States of America, 3 Louisiana Public Health Institute, New Orleans, Louisiana, United States of America, 4 Department of Population Medicine, Harvard Pilgrim Health Care Institute, Harvard Medical School, Boston, Massachusetts, United States of America, 5 Institute for Medicine and Public Health, Vanderbilt University Medical Center, Nashville, Tennessee, United States of America, 6 Health Outcomes and Biomedical Informatics, University of Florida Health, Gainesville, Florida, United States of America, 7 Utah COVID-19 Long Haulers, Salt Lake City, Utah, United States of America, 8 Patient Representative, Dallas, Georgia, United States of America

¶ Membership of the RECOVER Consortium is provided in the Acknowledgments.
* jav4003@med.cornell.edu

**Data Availability Statement:** There are legal and ethical restrictions on data sharing because the

## Abstract

### Importance

The frequency and characteristics of post-acute sequelae of SARS-CoV-2 infection (PASC) may vary by SARS-CoV-2 variant.

### Objective

To characterize PASC-related conditions among individuals likely infected by the ancestral strain in 2020 and individuals likely infected by the Delta variant in 2021.

### Design

Retrospective cohort study of electronic medical record data for approximately 27 million patients from March 1, 2020-November 30, 2021.

### Setting

Healthcare facilities in New York and Florida.

### Participants

Patients who were at least 20 years old and had diagnosis codes that included at least one SARS-CoV-2 viral test during the study period.

Institutional Review Board of Weill Cornell Medicine did not approve public data deposition. The data set used for this study constitutes sensitive patient information extracted from the electronic health records. Accordingly, it is subject to federal legislation that limits our ability to disclose it to the public, even after it has been subjected to deidentification techniques. To request the access of the de-identified minimal dataset underlying these findings, interested and qualified researchers should contact INSIGHT Clinical Research Network (https://insightcrn.org/) or Alexandra LaMar at all4008@med.cornell.edu.

**Funding:** This study is part of the NIH Researching COVID to Enhance Recovery (RECOVER) Initiative, which seeks to understand, treat, and prevent the post-acute sequelae of SARS-CoV-2 infection (PASC). This research was funded by the National Institutes of Health (NIH) Agreement OTA OT2HL161847 as part of the Researching COVID to Enhance Recovery (RECOVER) research program. NIH played a role in evaluating and developing the overall structure of the RECOVER research program, but not in the design and analysis of this specific study.

**Competing interests:** The authors have declared that no competing interests exist.

## Exposure

Laboratory-confirmed COVID-19 infection, classified by the most common variant prevalent in those regions at the time.

## Main outcome(s) and measure(s)

Relative risk (estimated by adjusted hazard ratio [aHR]) and absolute risk difference (estimated by adjusted excess burden) of new conditions, defined as new documentation of symptoms or diagnoses, in persons between 31–180 days after a positive COVID-19 test compared to persons without a COVID-19 test or diagnosis during the 31–180 days after the last negative test.

## Results

We analyzed data from 560,752 patients. The median age was 57 years; 60.3% were female, 20.0% non-Hispanic Black, and 19.6% Hispanic. During the study period, 57,616 patients had a positive SARS-CoV-2 test; 503,136 did not. For infections during the ancestral strain period, pulmonary fibrosis, edema (excess fluid), and inflammation had the largest aHR, comparing those with a positive test to those without a COVID-19 test or diagnosis (aHR 2.32 [95% CI 2.09 2.57]), and dyspnea (shortness of breath) carried the largest excess burden (47.6 more cases per 1,000 persons). For infections during the Delta period, pulmonary embolism had the largest aHR comparing those with a positive test to a negative test (aHR 2.18 [95% CI 1.57, 3.01]), and abdominal pain carried the largest excess burden (85.3 more cases per 1,000 persons).

## Conclusions and relevance

We documented a substantial relative risk of pulmonary embolism and a large absolute risk difference of abdomen-related symptoms after SARS-CoV-2 infection during the Delta variant period. As new SARS-CoV-2 variants emerge, researchers and clinicians should monitor patients for changing symptoms and conditions that develop after infection.

## Background

SARS-CoV-2 virus may cause persistent symptoms, exacerbations of existing conditions, or onset of new diseases in the weeks to months after initial infection [1]. These symptoms and conditions are generally referred to as "post-acute sequelae of SARS-CoV-2 infection" (PASC) in the medical literature and "long COVID" in the lay press, and defined as ongoing, relapsing, or new symptoms, or other health effects occurring after the acute phase of SARS-CoV-2 infection (i.e., present four or more weeks after the acute infection). Studies have produced markedly varying estimates of PASC, which may be due to differences in methods or due to true differences in the populations studied, severity of illness, viral genotype, or dose of virus that caused infection. One meta-analysis estimated that, globally, 49% of people report persistent symptoms 120 days after infection with an increased frequency in persons who are female or required hospitalization [2].

Since the emergence of SARS-CoV-2 in 2019, mutations in the viral nucleic acid sequence have changed the transmissibility, virulence, and immunogenicity of the virus [3]. The World

Health Organization (WHO) determines whether a new genotype has phenotypic characteristics that impact public health sufficiently to be classified as a variant of concern (VOC). In 2020, the US experienced a COVID-19 "wave" (generally considered a marked increase in infections, hospitalizations, and deaths) due to the ancestral strain and subsequent waves at the end of 2020 due to a mix of the ancestral strain and the alpha variant, then again in 2021 due to the Delta variant. Analyses suggest the Delta variant was more transmissible than the ancestral strain, but not necessarily more likely to cause severe illness and death [4].

PASC may result from some combination of persistent viral infection, an exaggerated immune response to initial infection, and tissue damage from the combination of initial infection and immune response [5]. It is, therefore, possible that the frequency or characteristics of PASC may vary depending on infection with different VOCs.

We analyzed data from a large database of electronic health records (EHR) in the United States to evaluate differences in PASC among those infected during the COVID-19 wave caused by the ancestral lineage in 2020 and those infected during the COVID-19 wave caused by the Delta variant in 2021.

## Methods

### Cohort enrollment and follow-up

We conducted a retrospective cohort study of approximately 27 million people receiving medical care in New York and Florida in the United States from March 1, 2020, to November 30, 2021. Patient records were obtained from two large clinical research networks within PCORnet, the National Patient Centered Clinical Research Network [6]: INSIGHT, which contains records from approximately 12 million persons who received services across five health systems (Albert Einstein School of Medicine/Montefiore Medical Center, Columbia University and Weill Cornell Medicine/New York-Presbyterian Hospital, lcahn School of Medicine/Mount Sinai Health System, and New York University School of Medicine/Langone Medical Center) in the New York City (NYC) metropolitan area [7], and OneFlorida+, which contains records from approximately 15 million persons receiving services across 13 health systems (University of Florida and UF Health, Florida State University, University of Miami and UHealth, Orlando Health System, AdventHealth, Tallahassee Memorial HealthCare, Tampa General Hospital, Bond Community Health Center Inc., Nicklaus Children's Hospital, CommunityHealth IT, University of South Florida and USF Health, University of Alabama at Birmingham, Emory University) in Florida [8]. Data elements from these databases are maintained on the PCORnet Common Data Model and mapped to the Observational Medical Outcomes Partnership Common Data Model for interoperability. Both networks receive data monthly from each in-network facility and create an integrated dataset of all encounters, diagnoses, procedures, medications, vitals, and social history.

In this analysis, patients were included if they were at least 20 years old and had at least one diagnosis code during a baseline period, and the follow-up period included at least one SARS-CoV-2 viral diagnostic test (antigen or molecular). The baseline period was defined as three years to seven days before the date of the first documented positive or negative SARS-CoV-2 test (referred to as the index date) for the infected group or the negative group, respectively. We required that patients in the negative group had all negative SARS-CoV-2 viral diagnostic tests and no COVID-19-related diagnoses. Requiring that patients had a least one diagnosis during this baseline period helped ensure that they were connected to the healthcare system and could have been diagnosed during baseline for relevant symptoms and conditions. The baseline period was defined as 3 years to 1 week before lab-confirmed SARS-CoV-2 infection and the follow-up period was defined as 31 to 180 days after the index date.

## COVID-19 Variants

Our dataset did not include data about the genotype of the virus that infected each person with COVID-19. Instead, we defined the beginning and end of each variant wave using COVID-19 test data from the networks [S1 Fig] and assumed that all infections occurring during those waves were attributable to the most common genotype prevalent in those regions at the time according to data from CDC [9]. We defined the ancestral strain period as March 1 -September 30, 2020, and the Delta variant period as June 1-November 30, 2021. For this analysis, we did not analyze the winter 2020–2021 period, because multiple genotypes were circulating [10].

## Data analysis

We investigated the following likely PASC categories based on prior analyses, including anemia, thromboembolism, pulmonary embolism, dementia, pulmonary fibrosis, edema, and inflammation, pressure ulcer, diabetes mellitus, malnutrition, fluid disorders, encephalopathy, abnormal heartbeat, chest pain, abdominal pain, constipation, joint pain, cognitive problems, headache, sleep disorders, dyspnea, acute pharyngitis, hair loss, edema, fever, malaise and fatigue [11]. In addition, we used ICD-10 diagnostic codes B948 (sequelae of other specified infectious and parasitic diseases) and U099 (post-COVID-19 condition, unspecified) to capture general or unspecified PASC conditions. Each condition was defined based on the Clinical Classifications Software Refined (CCSR) v2022.1. Codes that could not be attributed to COVID-19 were removed by clinicians.

We analyzed the risks of newly incident conditions, defined as new documentation of the above-mentioned PASC categories in the follow-up period that were not present in the baseline period. Specifically, we compared adjusted hazard ratios (aHR) and adjusted excess burdens of these events occurring 31–180 days after the index date, namely the follow-up period, between the SARS-CoV-2 positive group and the negative group. For each potential PASC condition, aHR was estimated by a Cox proportional hazard model, and the excess burden was defined as the difference in cumulative incidence per 1,000 patients in the positive group and negative group over the follow-up period. For example, an excess burden of 40 for symptom X indicates there were 40 more people per 1,000 with symptom X after COVID-19 infection compared with people not infected with COVID-19. We estimated cumulative incidence by the Aalen-Johansen model [12] considering death to be a competing risk for target outcomes. We adjusted for a wide range of baseline covariates by stabilized inverse propensity score re-weighting [13]. The standardized mean difference (SMD) was used to quantify the goodness-of-balance of covariates after reweighting. We considered SMD < 0.1 as being balanced in terms of each covariate and required all covariates to be balanced after re-weighting.

Both the aHR and excess burden calculations used the same covariates for adjustment. We summarized the baseline covariates in Table 1 and have included detailed descriptions in our previous studies [14, 17]. Briefly speaking, the baseline covariates included age, gender, race, ethnicity, and additional factors described here. The national-level area deprivation index (ADI) was used to assess the socioeconomic disadvantage of patients [15]. We imputed a missing ADI value with median ADI per site. Healthcare utilization was measured as the number of inpatient, outpatient, and emergency encounters (0, 1–2, 3–4, 5 or more visits for each encounter type). The Body Mass Index (BMI) was categorized according to WHO guidelines. We adopted a tailored list of the Elixhauser comorbidities and related drug categories (e.g., corticosteroid and immunosuppressant prescriptions) to capture comorbidities [16]. Patients were defined as having comorbidity if they had at least two corresponding diagnoses

**Table 1. Population characteristics of the lab-confirmed SARS-CoV-2 positive patients (cases) and SARS-CoV-2 negative patients (control) for the entire study period (March 2020 to November 2021), ancestral strain period (March 2020 to September 2020), and Delta variant period (June 2021 to November 2021).**

| Characteristics | Entire Study Period | | Ancestral Strain Period | | Delta Period | |
|---|---|---|---|---|---|---|
| | Cases | Control | Cases | Control | Cases | Control |
| N | 57,616 | 503,136 | 19,943 | 215,842 | 8,097 | 52,717 |
| **Median age (IQR)—yrs** | 53 (36–66) | 57 (40–69) | 53 (36–66) | 57 (41–69) | 48 (33–63) | 57 (40–70) |
| **Age group—no. (%)** | | | | | | |
| 20-<40 years | 17,035 (29.6) | 119,689 (23.8) | 5,928 (29.7) | 50,855 (23.6) | 2,972 (36.7) | 13,119 (24.9) |
| 40-<55 years | 13,448 (23.3) | 107,868 (21.4) | 4,680 (23.5) | 46,058 (21.3) | 2,003 (24.7) | 10,579 (20.1) |
| 55-<65 years | 11,001 (19.1) | 103,503 (20.6) | 3,832 (19.2) | 45,262 (21.0) | 1,380 (17.0) | 10,219 (19.4) |
| 65-<75 years | 8,641 (15.0) | 97,461 (19.4) | 2,957 (14.8) | 41,961 (19.4) | 997 (12.3) | 9,993 (19.0) |
| 75+ years | 7,491 (13.0) | 74,615 (14.8) | 2,546 (12.8) | 31,706 (14.7) | 745 (9.2) | 8,807 (16.7) |
| **Sex—no. (%)** | | | | | | |
| Female | 34,690 (60.2) | 303,693 (60.4) | 11,907 (59.7) | 129,090 (59.8) | 5,126 (63.3) | 32,304 (61.3) |
| Male | 22,921 (39.8) | 199,394 (39.6) | 8,034 (40.3) | 86,732 (40.2) | 2,970 (36.7) | 20,408 (38.7) |
| **Race—no. (%)** | | | | | | |
| Asian | 2,011 (3.5) | 20,351 (4.0) | 771 (3.9) | 9,470 (4.4) | 123 (1.5) | 1,795 (3.4) |
| Black | 14,295 (24.8) | 97,662 (19.4) | 5,589 (28.0) | 40,850 (18.9) | 2,253 (27.8) | 11,049 (21.0) |
| White | 23,631 (41.0) | 245,033 (48.7) | 7,036 (35.3) | 107,251 (49.7) | 4,143 (51.2) | 26,341 (50.0) |
| Other | 13,574 (23.6) | 99,544 (19.8) | 5,124 (25.7) | 42,773 (19.8) | 1,237 (15.3) | 9,307 (17.7) |
| Missing | 4,105 (7.1) | 40,546 (8.1) | 1,423 (7.1) | 15,498 (7.2) | 341 (4.2) | 4,225 (8.0) |
| **Ethnic group—no. (%)** | | | | | | |
| Hispanic: Yes | 15,158 (26.3) | 95,006 (18.9) | 5,482 (27.5) | 38,199 (17.7) | 1,701 (21.0) | 10,221 (19.4) |
| Hispanic: No | 35,636 (61.9) | 336,494 (66.9) | 12,152 (60.9) | 149,272 (69.2) | 5,506 (68.0) | 34,280 (65.0) |
| Hispanic: Other/Missing | 6,822 (11.8) | 71,636 (14.2) | 2,309 (11.6) | 28,371 (13.1) | 890 (11.0) | 8,216 (15.6) |
| **Median area deprivation index (IQR)—rank** | 25 (12–53) | 24 (10–48) | 25 (12–52) | 24 (10–48) | 50 (27–70) | 30 (15–58) |
| **Follow-up days (IQR)** | 246 (133–386) | 263 (136–394) | 471 (317–539) | 403 (255–479) | 92 (59–128) | 86 (57–121) |
| **Death in follow-up (%)** | 724 (1.3) | 4,676 (0.9) | 275 (1.4) | 2,665 (1.2) | 102 (1.3) | 380 (0.7) |
| **No. of hospital visits in the past 3 yrs—no. (%)** | | | | | | |
| Inpatient 0 | 38,555 (66.9) | 391,264 (77.8) | 13,543 (67.9) | 156,773 (72.6) | 4,927 (60.8) | 43,269 (82.1) |
| Inpatient 1–2 | 11,419 (19.8) | 70,955 (14.1) | 3,704 (18.6) | 35,211 (16.3) | 1,741 (21.5) | 6,672 (12.7) |
| Inpatient >= 3 | 7,642 (13.3) | 40,917 (8.1) | 2,696 (13.5) | 23,858 (11.1) | 1,429 (17.6) | 2,776 (5.3) |
| Outpatient 0 | 5,327 (9.2) | 28,119 (5.6) | 1,829 (9.2) | 11,857 (5.5) | 1,249 (15.4) | 3,846 (7.3) |
| Outpatient 1–2 | 5,944 (10.3) | 50,669 (10.1) | 2,263 (11.3) | 22,269 (10.3) | 849 (10.5) | 4,909 (9.3) |
| Outpatient >= 3 | 46,345 (80.4) | 424,348 (84.3) | 15,851 (79.5) | 181,716 (84.2) | 5,999 (74.1) | 43,962 (83.4) |
| Emergency 0 | 29,234 (50.7) | 335,023 (66.6) | 10,174 (51.0) | 139,775 (64.8) | 3,279 (40.5) | 34,423 (65.3) |
| Emergency 1–2 | 14,669 (25.5) | 97,067 (19.3) | 4,959 (24.9) | 41,575 (19.3) | 2,050 (25.3) | 11,031 (20.9) |
| Emergency >= 3 | 13,713 (23.8) | 71,046 (14.1) | 4,810 (24.1) | 34,492 (16.0) | 2,768 (34.2) | 7,263 (13.8) |
| **BMI (IQR)** | 28 (23–33) | 26 (21–31) | 27 (23–33) | 26 (21–31) | 29 (24–34) | 26 (20–31) |
| BMI: <18.5 under weight | 6,848 (11.9) | 93,344 (18.6) | 2,425 (12.2) | 35,439 (16.4) | 537 (6.6) | 11,013 (20.9) |
| BMI: 18.5-<25 normal weight | 10,304 (17.9) | 103,463 (20.6) | 3,586 (18.0) | 46,361 (21.5) | 1,405 (17.4) | 10,271 (19.5) |
| BMI: 25-<30 overweight | 12,936 (22.5) | 116,673 (23.2) | 4,376 (21.9) | 50,686 (23.5) | 1,794 (22.2) | 11,924 (22.6) |
| BMI: >= 30 obese | 18,248 (31.7) | 137,366 (27.3) | 6,224 (31.2) | 58,810 (27.2) | 2,826 (34.9) | 14,672 (27.8) |
| BMI: missing | 9,280 (16.1) | 52,290 (10.4) | 3,332 (16.7) | 24,546 (11.4) | 1,535 (19.0) | 4,837 (9.2) |
| **Pre-existing conditions—no. (%)** | | | | | | |
| Alcohol Abuse | 1,741 (3.0) | 15,236 (3.0) | 602 (3.0) | 7,547 (3.5) | 284 (3.5) | 1,480 (2.8) |
| Anemia | 8,933 (15.5) | 53,673 (10.7) | 3,188 (16.0) | 26,112 (12.1) | 1,369 (16.9) | 5,314 (10.1) |
| Arrythmia | 8,425 (14.6) | 57,916 (11.5) | 3,075 (15.4) | 28,058 (13.0) | 1,042 (12.9) | 5,782 (11.0) |
| Asthma | 6,546 (11.4) | 39,531 (7.9) | 2,115 (10.6) | 17,885 (8.3) | 1,000 (12.4) | 3,888 (7.4) |

*(Continued)*

**Table 1.** (Continued)

| Characteristics | Entire Study Period | | Ancestral Strain Period | | Delta Period | |
|---|---|---|---|---|---|---|
| | Cases | Control | Cases | Control | Cases | Control |
| Autism | 136 (0.2) | 866 (0.2) | 58 (0.3) | 359 (0.2) | 23 (0.3) | 104 (0.2) |
| Cancer | 5,584 (9.7) | 58,310 (11.6) | 1,981 (9.9) | 29,302 (13.6) | 691 (8.5) | 5,646 (10.7) |
| Chronic Kidney Disease | 8,383 (14.5) | 47,376 (9.4) | 3,016 (15.1) | 22,614 (10.5) | 1,086 (13.4) | 5,209 (9.9) |
| Chronic Pulmonary Disorders | 10,644 (18.5) | 70,105 (13.9) | 3,535 (17.7) | 33,185 (15.4) | 1,596 (19.7) | 6,760 (12.8) |
| Cirrhosis | 950 (1.6) | 7,953 (1.6) | 295 (1.5) | 3,958 (1.8) | 147 (1.8) | 804 (1.5) |
| Coagulopathy | 3,652 (6.3) | 17,039 (3.4) | 1,319 (6.6) | 8,880 (4.1) | 466 (5.8) | 1,595 (3.0) |
| Congestive Heart Failure | 6,602 (11.5) | 42,007 (8.3) | 2,310 (11.6) | 20,811 (9.6) | 944 (11.7) | 4,322 (8.2) |
| COPD | 3,326 (5.8) | 23,118 (4.6) | 1,180 (5.9) | 12,019 (5.6) | 478 (5.9) | 2,058 (3.9) |
| Coronary Artery Disease | 7,400 (12.8) | 52,920 (10.5) | 2,522 (12.6) | 25,232 (11.7) | 919 (11.3) | 5,296 (10.0) |
| Cystic Fibrosis | 29 (0.1) | 307 (0.1) | 4 (0.0) | 174 (0.1) | 9 (0.1) | 24 (0.0) |
| Dementia | 2,313 (4.0) | 9,430 (1.9) | 997 (5.0) | 4,526 (2.1) | 205 (2.5) | 970 (1.8) |
| Diabetes Type 1 | 829 (1.4) | 4,676 (0.9) | 291 (1.5) | 2,241 (1.0) | 132 (1.6) | 440 (0.8) |
| Diabetes Type 2 | 13,004 (22.6) | 77,745 (15.5) | 4,694 (23.5) | 34,920 (16.2) | 1,635 (20.2) | 8,376 (15.9) |
| Down's Syndrome | 65 (0.1) | 251 (0.0) | 30 (0.2) | 94 (0.0) | 14 (0.2) | 28 (0.1) |
| End Stage Renal Disease on Dialysis | 2,468 (4.3) | 10,795 (2.1) | 1,012 (5.1) | 5,611 (2.6) | 320 (4.0) | 1,089 (2.1) |
| Hemiplegia | 785 (1.4) | 4,292 (0.9) | 313 (1.6) | 2,216 (1.0) | 86 (1.1) | 385 (0.7) |
| HIV | 774 (1.3) | 7,074 (1.4) | 278 (1.4) | 3,060 (1.4) | 95 (1.2) | 695 (1.3) |
| Hypertension | 23,650 (41.0) | 169,604 (33.7) | 8,041 (40.3) | 75,509 (35.0) | 3,226 (39.8) | 18,288 (34.7) |
| Inflammatory Bowel Disorder | 618 (1.1) | 7,693 (1.5) | 199 (1.0) | 3,602 (1.7) | 100 (1.2) | 655 (1.2) |
| Lupus or Systemic Lupus Erythematosus | 547 (0.9) | 3,430 (0.7) | 157 (0.8) | 1,617 (0.7) | 96 (1.2) | 346 (0.7) |
| Mental Health Disorders | 7,665 (13.3) | 54,170 (10.8) | 2,615 (13.1) | 25,421 (11.8) | 1,252 (15.5) | 5,724 (10.9) |
| Multiple Sclerosis | 296 (0.5) | 2,003 (0.4) | 71 (0.4) | 960 (0.4) | 45 (0.6) | 198 (0.4) |
| Other Substance Abuse | 4,919 (8.5) | 45,367 (9.0) | 1,509 (7.6) | 21,651 (10.0) | 1,109 (13.7) | 4,850 (9.2) |
| Parkinson's Disease | 365 (0.6) | 3,347 (0.7) | 141 (0.7) | 1,474 (0.7) | 33 (0.4) | 411 (0.8) |
| Peripheral vascular disorders | 4,257 (7.4) | 30,330 (6.0) | 1,521 (7.6) | 14,727 (6.8) | 527 (6.5) | 3,076 (5.8) |
| Pregnant | 3,112 (5.4) | 26,072 (5.2) | 1,085 (5.4) | 10,982 (5.1) | 678 (8.4) | 3,432 (6.5) |
| Pulmonary Circulation Disorder | 2,272 (3.9) | 11,967 (2.4) | 796 (4.0) | 6,341 (2.9) | 329 (4.1) | 1,110 (2.1) |
| Rheumatoid Arthritis | 997 (1.7) | 6,870 (1.4) | 347 (1.7) | 3,176 (1.5) | 131 (1.6) | 666 (1.3) |
| Seizure/Epilepsy | 1,585 (2.8) | 9,720 (1.9) | 628 (3.1) | 4,814 (2.2) | 244 (3.0) | 974 (1.8) |
| Severe Obesity (BMI>= 40 kg/m2) | 5,298 (9.2) | 31,619 (6.3) | 1,803 (9.0) | 13,859 (6.4) | 882 (10.9) | 3,477 (6.6) |
| Sickle Cell | 482 (0.8) | 2,644 (0.5) | 164 (0.8) | 1,248 (0.6) | 90 (1.1) | 299 (0.6) |
| Weight Loss | 2,728 (4.7) | 14,932 (3.0) | 985 (4.9) | 8,111 (3.8) | 385 (4.8) | 1,283 (2.4) |
| Corticosteroids Drugs | 9,252 (16.1) | 56,698 (11.3) | 3,076 (15.4) | 27,730 (12.8) | 1,467 (18.1) | 5,271 (10.0) |
| Immunosuppressant Drugs | 3,123 (5.4) | 18,042 (3.6) | 1,054 (5.3) | 8,888 (4.1) | 432 (5.3) | 1,713 (3.2) |

a. The lab-confirmed SARS-CoV-2 positive and negative patients were identified by polymerase chain reaction (PCR) test or antigen test. Negative patients were further required no documented COVID-19 related diagnoses at any time. IQR denotes inter-quartile range. Percentage may not sum up to 100 because of rounding.

c. Coexisting conditions existed if two records in the 3-years prior to index event. See detailed phenotyping codes in the appendix. SLE: Systemic Lupus Erythematosus; COPD: Chronic obstructive pulmonary disease.

documented during the baseline period. The clinical features were collected during the baseline period, namely 3 years to 1 week before lab-confirmed SARS-CoV-2 infection.

We reported PASC conditions if they had: adjusted hazard ratio > 1; P-value $<3.6 \times 10^{-4}$ (corrected by the Bonferroni method to control for false discovery) in multiple test settings; and at least 100 patients with the condition. We reported adjusted hazard ratios with a 95% confidence interval. We used Python 3.9, python package lifelines-0.2666 for survival analysis,

and scikit-learn-0.2318 for machine learning models. The code is available at https://github.com/calvin-zcx/pasc_phenotype.

### Ethical review

The use of the INSIGHT data was approved by the Institutional Review Board (IRB) of Weill Cornell Medicine following NIH protocol 21-10-95-380 with protocol title: Adult PCORnet-PASC Response to the Proposed Revised Milestones for the PASC EHR/ORWD Teams (RECOVER). The use of the OneFlorida+ data for this study was approved under the University of Florida IRB number IRB202001831. The IRBs waived informed consent for this observational cohort study that involves a limited, but not de-identified, data set.

## Results

### Demographics

Our final dataset included data from 560,752 patients [Fig 1]. The median age of patients was 57 years (interquartile range [IQR] 40–69), and 338,383 (60.3%) were female [Table 1]. According to EHR data, 268,664 (47.9%) patients identified as non-Hispanic white, 111,957 (20.0%) identified as non-Hispanic Black, 110,164 (19.6%) identified as Hispanic, 22,362 (4.0%) identified as Asian/Pacific Islander, and 44,651 (8.0%) had missing race and ethnicity. The median Body Mass Index (BMI) was 26 (IQR 21, 32).

For the overall study period, 57,616 patients had a positive molecular or antigen test (99.8% were molecular) for SARS-CoV-2 (19,943 during the ancestral strain period, 8,097 during Delta variant period, 29,576 outside of these two periods), and 503,136 did not (215,842 during ancestral strain period, 52,717 during Delta variant period, 234,577 outside of these two periods). We observed that cases in the Delta variant period had a median age of 48 (IQR 33, 63), and cases in the ancestral strain period had a median age of 53 ([IQR 36, 66]). The proportion of female cases in the Delta variant period was 63.3% compared with 59.7% in the ancestral strain period.

Further information about co-morbid conditions of both cases and controls over the different periods is in Table 1.

### PASC associated with ancestral strain period

We found an increased risk of multiple symptoms and conditions from ancestral strain infections. As shown in Fig 2, the largest adjusted hazard ratios, comparing patients with positive vs. negative were for: pulmonary fibrosis, edema, and inflammation (2.32 [95% CI 2.09 2.57]), hair loss (2.10[95% CI 1.74 2.52]), pressure ulcers (1.98 [95% CI 1.69 2.32]), pulmonary embolism (1.65 [95% CI 1.36 2.99]), dyspnea (1.58 [95% CI 1.45 1.69]), dementia (1.50 [95% CI 1.28 1.74]). As shown in Fig 3, the largest excess burdens were for: dyspnea (47.6 more cases per 1000 persons), pulmonary fibrosis, edema, and inflammation (21.5), malaise and fatigue (18.2), edema (17.6), chest pain (16.9), abnormal heartbeat (15.4), cognitive problems (12.8), and joint pain (11.5).

### PASC associated with Delta variant period

The spectrum of PASC-related symptoms and conditions from Delta infections varied from ancestral strain infections. For the Delta variant period, the largest adjusted hazard ratios, comparing patients with positive vs. negative tests [Fig 2] were for: pulmonary embolism (2.18 [95% CI 1.57 3.01]), hair loss (2.07 [95% CI 1.31 3.27]), pulmonary fibrosis, edema, and inflammation (1.99 [95% CI 1.54 2.57]), pressure ulcers (1.64 [95% CI 1.15 2.35]), acute

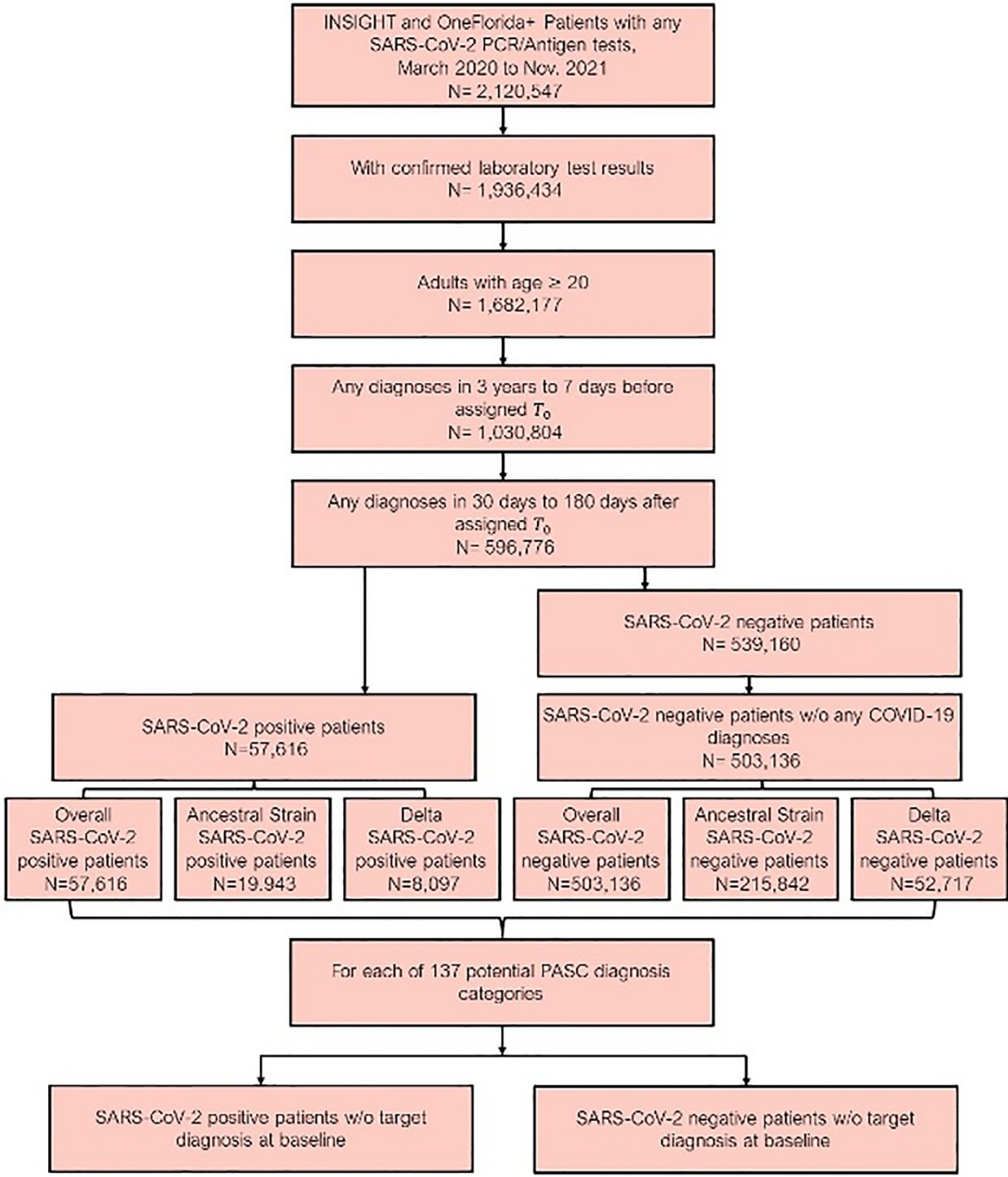

**Fig 1. Patients included from both the INSIGHT and OneFlorida+, March 2020 to November 2021.** The post-acute sequelae of SARS-CoV-2 infection (PASC) outcomes were ascertained from day 30 after the SARS-CoV-2 infection (index date) and the adjusted risk was computed 180 days after the index date.

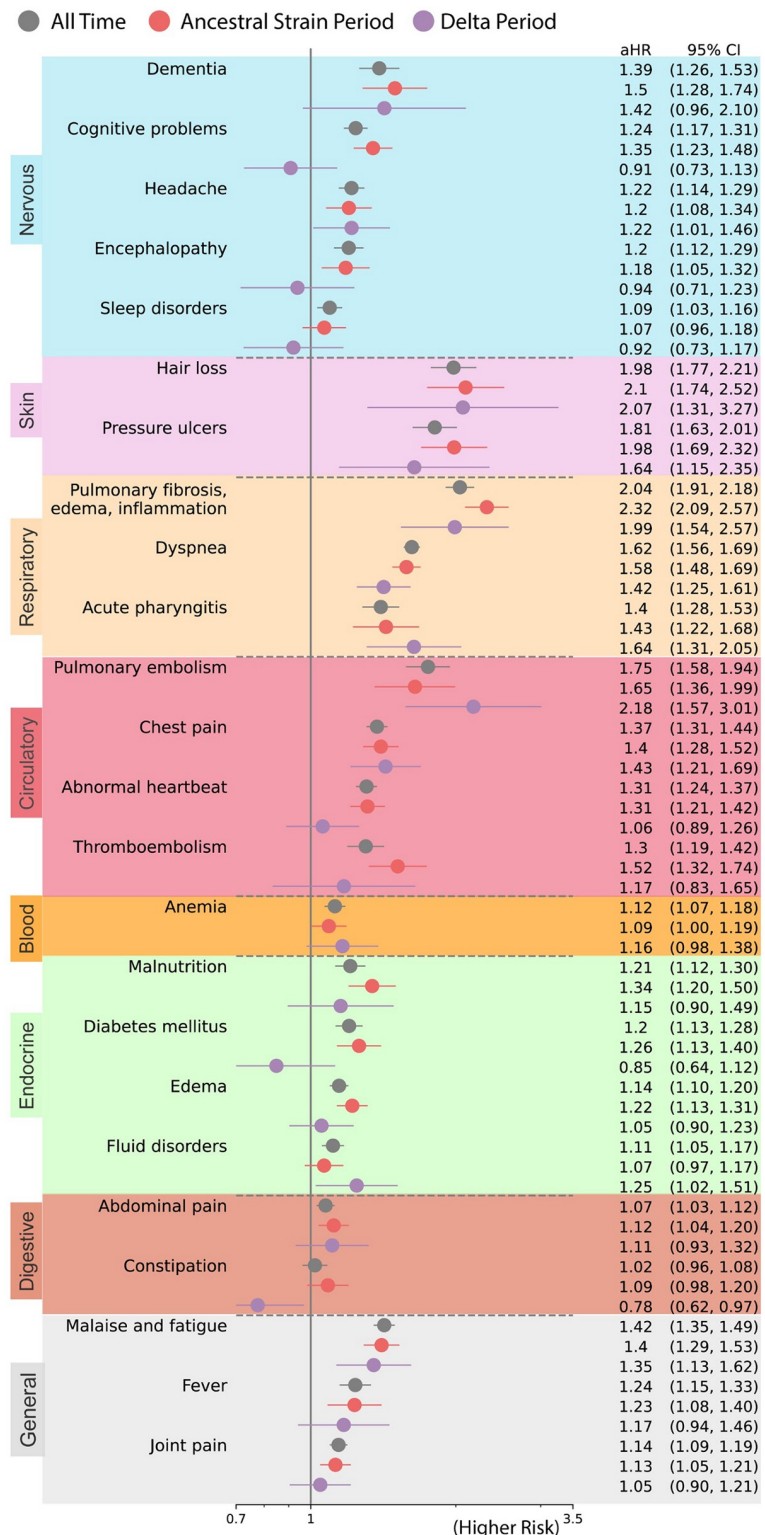

**Fig 2. Adjusted hazard ratios of likely incident PASC conditions in all time (March 2020 to November 2021) versus ancestral strain period (March 2020 to September 2020) versus Delta variant period (June 2021 to November 2021).** Sequelae outcomes were ascertained from day 30 after the SARS-CoV-2 infection and the adjusted hazard ratio was computed at 180 days after the SARS-CoV-2 infection. The color panels from top to bottom represent different organ systems, including the nervous system or mental disorders, skin, respiratory system, circulatory system, blood, endocrine and metabolic, digestive system, and other signs.

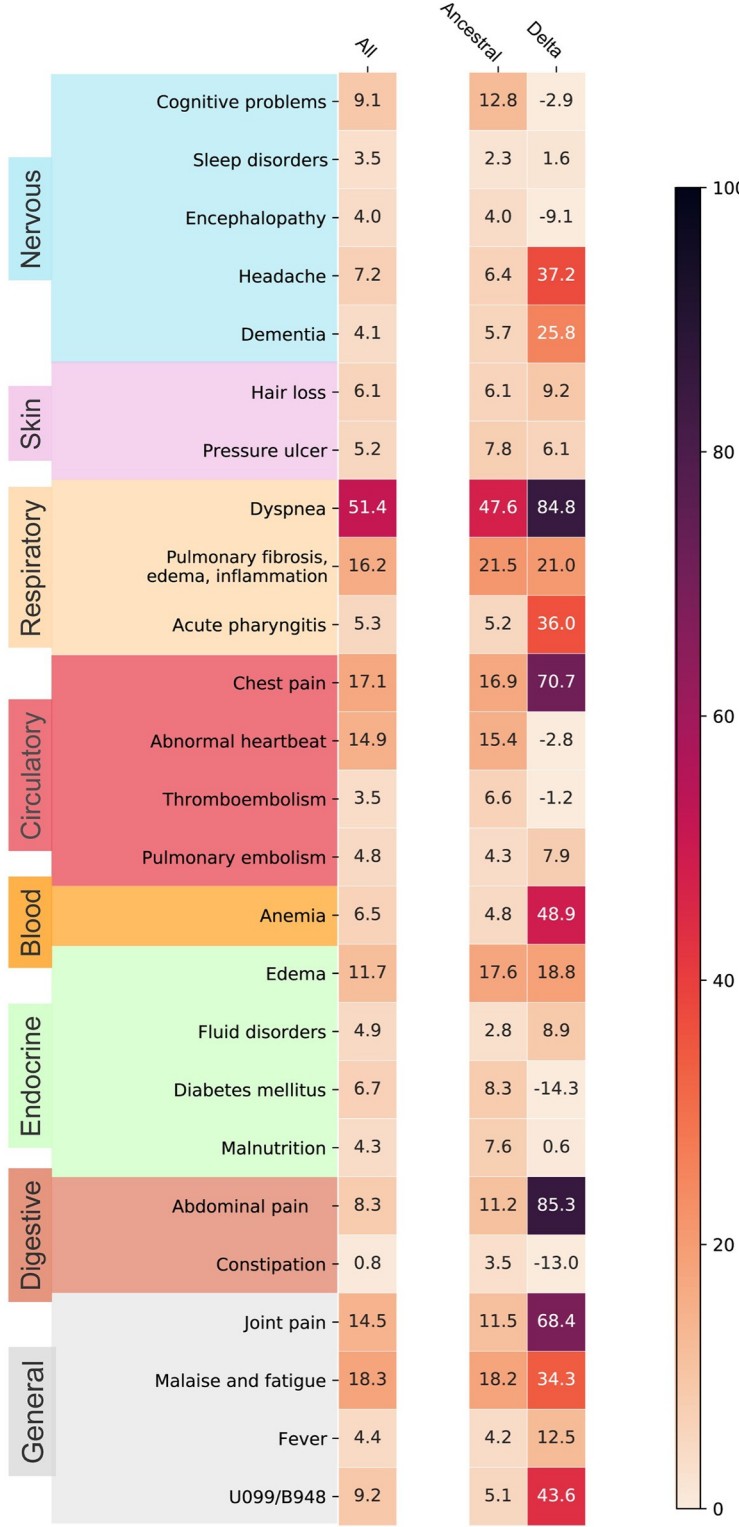

**Fig 3. Adjusted excess burden per 1,000 patients of potential incident PASC conditions for the entire study period (March 2020-November 2021), ancestral strain period (March 2020-September 2020), and Delta variant period (June 2021-November 2021).** The excess burdens were represented by a heatmap with deeper colors representing more significant burdens. The color panels from top to bottom represent different organ systems, including the nervous system or mental disorders, skin, respiratory system, circulatory system, blood, endocrine and metabolic, digestive system, and other signs.

pharyngitis (1.64 [95% CI 1.31 2.05]), chest pain (1.43 [95% CI 1.21 1.69]). As shown in Fig 3, the largest excess burdens were for: abdominal pain (85.3 more cases per 1000 persons), dyspnea (84.8), chest pain (70.7), joint pain (68.4), anemia (48.9), headache (37.2), acute pharyngitis (36.0), malaise and fatigue (34.3), dementia (25.8), pulmonary fibrosis, edema, and inflammation (21.0), and edema (18.8).

Similar patterns were observed in the region-specified sensitivity analyses as shown in S2 Fig and S1 Table.

## Discussion

In this study of more than half a million patients, we documented an increased burden of abdomen-related conditions during the Delta variant period and a substantial burden of post-acute pulmonary-related conditions in people infected with COVID-19 during both the ancestral strain and Delta variant periods.

Our statistical approach considered both the relative risk of a new condition, expressed as a hazard ratio, and the risk difference, expressed as an excess burden. The risk difference provides a measure of public health impact because it estimates how much each symptom or condition could potentially be reduced if patients had not been infected with COVID-19 ancestral or Delta variants. We estimate that, for every 1,000 patients, there were an additional 85 persons with abdominal pain after Delta variant infection than in those without documentation of Delta variant infection. The excess burden was, notably, not found during the ancestral strain period. Since the beginning of the pandemic, clinicians have noted that patients may present with pronounced gastrointestinal symptoms, possibly due to direct infection, alteration of the gut microbiome, or enhanced immune response [17, 18]. Whether these same mechanisms explain post-acute sequelae is unclear. Further research is needed to understand the long-term prognosis of PASC-related abdominal pain and to quantify its excess burden, given that international experts did not include gastrointestinal conditions in its core outcome set for evaluating patients [19].

The excess burden of pulmonary-related conditions was large and markedly increased for the Delta variant period. We found a 78% increase in post-acute dyspnea from the ancestral strain to Delta variant periods (47.6 additional persons vs. 84.8 additional persons), even though the excess burden of diagnosed conditions that could explain dyspnea (pulmonary fibrosis, edema, and inflammation) was similar (21.5 vs. 21.0) during both periods. We do not know whether the increase in burden during the Delta variant period represents a specific change in the virus or some other factor, such as the types of persons infected, the dose, duration, or route of exposure, or increased awareness and documentation by providers. In the absence of hypoxemia, no current treatment exists for persistent dyspnea, although novel strategies to help patients, such as breathlessness training, are being evaluated [20]. Notably, the largest relative risk was associated with pulmonary embolism, a well-documented COVID-19 complication that could also cause persistent dyspnea [21].

Our study is subject to important limitations. First, we may have misclassified patients as not infected with COVID-19, because a test was never performed or not recorded in the in-network facilities. This may have been more likely during the follow-up of the post-Delta wave, as the follow-up period overlapped with the increasing availability of home testing. Such misclassification would likely lead us to underestimate the prevalence and relative risk of PASC, particularly during the first wave when diagnostic testing was less widely available. It is also possible that persons who did not test positive had other infections, which could produce illnesses similar to PASC [22]. Further misclassification could have occurred, because we assumed all infections during a wave were attributable to the most prevalent variant circulating

at that time or because patients suffered viral or bacterial co-infections at the same time as COVID-19 [23–26]. Second, we were not able to obtain vaccination information, which is important for further research given our finding that the burden of PASC was high during the Delta wave, despite the widespread availability of vaccines at that time [27]. Third, the number of cases during the Delta variant period was substantially higher in Florida (S1 Fig and S1 Table), and, for several conditions, the magnitude of excess burden varied between NY and Florida (S2 Fig), but our major conclusions still hold. Fourth, while we adjusted for many characteristics, other unmeasured factors or missing values (e.g., missing ADI value, smoking status, etc.) could explain differences. We examined PASC in patients who were still alive beyond their acute infection period, namely alive beyond 30 days, and we modeled death in the follow-up period (Table 1) as a competing risk in our analyses. However, different mortality rates in different periods might also contribute to the observed PASC differences. Finally, some conditions, such as pressure ulcers, may be attributable to prolonged hospitalization, rather than infection itself.

In conclusion, we found that conditions associated with PASC vary by viral variant. As the virus continues to evolve new variants rapidly, researchers and clinicians need to monitor for changing symptoms and conditions associated with COVID-19 infection. From the perspective of patients, physicians need to be aware that PASC may present differently in the future as new variants emerge and that treatment efficacy may vary by the variant that initially caused PASC.

## Supporting information

**S1 Fig. Lab-confirmed new SARS-CoV-2 cases per 10,000 patients with different variants of concerns in the INSIGHT and OneFlorida+ cohorts.**
(TIFF)

**S2 Fig. Adjusted excess burdens per 1,000 patients of potential incident PASC conditions in all time (March 2020 to November 2021) versus ancestral strain period (March 2020 to September 2020) versus Delta variant period (June 2021 to November 2021), stratified by different regions (NYC vs. Florida).** The sequelae outcomes were ascertained from day 30 after the SARS-CoV-2 infection and the adjusted hazard ratio were computed 180 days after the SARS-CoV-2 infection.
(TIFF)

**S1 Table. Number of patients in all time (March 2020 to November 2021) versus ancestral strain period (March 2020 to September 2020) versus Delta variant period (June 2021 to November 2021), stratified by different regions, NYC Insight vs. Florida OneFlorida.**
(DOCX)

## Acknowledgments

Members of the RECOVER Consortium include: Parsa Mirhaji, Albert Einstein College of Medicine; Ravi Jhaveri, Ann & Robert H. Lurie Children's Hospital of Chicago; Chris Forrest, Children's Hospital of Philadelphia; Hiroki Morizono, Children's National Medical Center; Nathan Pajor, Cincinnati Children's; Soumitra Sengupta, Columbia University; Schuyler Jones, Duke University/Health System; Benjamin D. Horne, Intermountain Healthcare; Tom Carton, LPHI; Bradley Taylor, Medical College of Wisconsin; Leslie Lenert, Medical University of South Carolina; Abu Mosa, University of Missouri; Carol Horowitz, Icahn School of Medicine at Mount Sinai; Kelly Kelleher, Nationwide Children's Hospital; H Timothy Bunnell, Nemours; David Liebovitz, Northwestern University; Saul Blecker, NYU Langone Health; Marion Sills, OCHIN, Inc.; Dan Fort, Ochsner Health; William Hogan, University of Florida;

Asa Oxner, University of South Florida; Rishi Kamaleswaran, Emory; Nick Tsinoremas, University of Miami; Daria Salyakina, Nicklaus Children's Hospital; Cynthia Chuang, Penn State U College of Medicine; Dimitri Christakis, Seattle Children's; Anuradha Paranjape, Temple University; Soledad Fernandez, The Ohio State University; Susan Kim, University of California San Fransisco; Elizabeth Chrischilles, University of Iowa; David Williams, University of Michigan; Carol Geary, University of Nebraska Medical Center; Suresh Srinivasan, Jonathan Arnold; Michael Bechich, University of Pittsburgh; Mollie Cummins, University of Utah; Lindsay Cowell, UT Southwestern Medical Center; Yacob Tedla, Vanderbilt University Medical Center; Stephen Downs, Wake Forest University Health Sciences; Rainu Kaushal, Weill Cornell Medicine (corresponding author); Alka Khaitan, Indiana University; Alan Schroeder, Stanford; Suchitra Rao, Univ. Colorado- Denver; Jyotsna Fuloria, University Medical Center New Orleans; Jason Block, Harvard Pilgram; Daria Salyakina, Nicklaus Children's Hospital.

## Statements and acknowledgements

1. Authorship has been determined by ICJME recommendation

2. Disclosures to be obtained at time of submission

3. The content is solely the responsibility of the authors and does not necessarily represent the official views of the RECOVER Program or funders

4. We would like to thank the National Community Engagement Group (NCEG), all patient, caregiver and community Representatives, and all the participants enrolled in the RECOVER Initiative.

## Author Contributions

**Conceptualization:** Jay K. Varma, Thomas W. Carton, Jason P. Block, Dhruv J. Khullar, Yongkang Zhang, Mark G. Weiner, Zhenxing Xu, Fei Wang, Rainu Kaushal.

**Data curation:** Chengxi Zang, Yongkang Zhang, Kristin Lyman, Jiang Bian, Jie Xu, Elizabeth A. Shenkman, Fei Wang.

**Formal analysis:** Jay K. Varma, Chengxi Zang, Yongkang Zhang, Fei Wang.

**Funding acquisition:** Thomas W. Carton, Rainu Kaushal.

**Methodology:** Jay K. Varma, Chengxi Zang, Jason P. Block, Yongkang Zhang, Mark G. Weiner, Russell L. Rothman, Edward J. Schenck, Jie Xu, Fei Wang, Rainu Kaushal.

**Project administration:** Jay K. Varma, Thomas W. Carton, Fei Wang, Rainu Kaushal.

**Resources:** Thomas W. Carton, Rainu Kaushal.

**Supervision:** Jay K. Varma, Fei Wang, Rainu Kaushal.

**Visualization:** Fei Wang.

**Writing – original draft:** Jay K. Varma.

**Writing – review & editing:** Jay K. Varma, Chengxi Zang, Thomas W. Carton, Jason P. Block, Dhruv J. Khullar, Yongkang Zhang, Mark G. Weiner, Russell L. Rothman, Edward J. Schenck, Zhenxing Xu, Kristin Lyman, Jiang Bian, Jie Xu, Elizabeth A. Shenkman, Christine Maughan, Leah Castro-Baucom, Lisa O'Brien, Fei Wang, Rainu Kaushal.

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
