## [Decision Letter · Decision Letter 0]

17 Aug 2023

PONE-D-23-04388Excess burden of respiratory and abdominal conditions following COVID-19 infections during the ancestral and Delta variant periods in the United States: An EHR-based cohort study from the RECOVER ProgramPLOS ONE

Dear Dr. Jay K Varma, 

Thank you for submitting your manuscript to PLOS ONE. After careful consideration, we feel that it has merit but does not fully meet PLOS ONE’s publication criteria as it currently stands. Therefore, we invite you to submit a revised version of the manuscript that addresses the points raised during the review process.

We look forward to receiving your revised manuscript.

Kind regards,

Roberto Scendoni

Academic Editor

PLOS ONE

Journal Requirements:

"This research was funded by the National Institutes of Health (NIH) Agreement OTA OT2HL161847 as part of the Researching COVID to Enhance Recovery (RECOVER) research program."

"This study is part of the NIH Researching COVID to Enhance Recovery (RECOVER) Initiative, which seeks to understand, treat, and prevent the post-acute sequelae of SARS-CoV-2 infection (PASC). This research was funded by the National Institutes of Health (NIH) Agreement OTA OT2HL161847 as part of the Researching COVID to Enhance Recovery (RECOVER) research program. NIH played a role in evaluating and developing the overall structure of the RECOVER research program, but not in the design and analysis of this specific study."

4. One of the noted authors is a group or consortium "the RECOVER Consortium". In addition to naming the author group, please list the individual authors and affiliations within this group in the acknowledgments section of your manuscript. Please also indicate clearly a lead author for this group along with a contact email address

Reviewers' comments:

Reviewer's Responses to Questions

**Comments to the Author**

1. Is the manuscript technically sound, and do the data support the conclusions?

Reviewer #1: Yes

Reviewer #2: Yes

2. Has the statistical analysis been performed appropriately and rigorously? 

Reviewer #1: Yes

Reviewer #2: Yes

3. Have the authors made all data underlying the findings in their manuscript fully available?

Reviewer #1: Yes

Reviewer #2: No

4. Is the manuscript presented in an intelligible fashion and written in standard English?

Reviewer #1: Yes

Reviewer #2: Yes

5. Review Comments to the Author

Reviewer #1: Thank you for the opportunity to review this manuscript. It is generally well written and clear, so my comments focus primarily on a few methodological questions and presentation.

In the methods, the authors state, “Patients were defined as having a comorbidity… 2 corresponding diagnoses documented during the baseline period.” I believe that I read earlier that the baseline period was as short as 7 days. Is it possible to have had 2 corresponding diagnoses during that short a time period? This is minor, but it does not make sense to me. Also, earlier in that paragraph “inpatients” should be “inpatient.”

Which covariates from Table 1 were included in the hazards ratio calculations? These should be stated in the methods.

The excess burden is quite different between the two time periods. Is it possible that more people died during the ancestral period, thus did not develop PASC? Were they included if they subsequently died? Was death considered a PASC? In any event, it seems that these differences in the magnitude of excess burden bears more discussion.

Figure 1 Legend: “Patient” should be “Patients,” suggest adding “(index date)” after “SARS-CoV-2.” The text about measurement dates following the title should be included in the methods.

Figure 1: There seem to be connector lines missing below the SARS cases and controls boxes and individual connector lines missing below the 6 boxes on the next row.

Figure 2: What is the significance of the dotted lines?

Figure 3: This figure is pretty, but a bit confusing. I do not believe that the figure of the person is necessary and should come out. You could more clearly present the data by adding a column to the left of the “all” column that indicates the system for each group of symptoms/conditions and also remove the color from those boxes. It might be helpful to state that the figure is a heat map and indicate that the “thermometer” is the legend.

I did not see reference to Supplemental Fig 2 in the text. Did I miss it?

Reviewer #2: I thank you the authors for the work they have done. I think the manuscript touches upon an important topic in medicine. The large sample makes the analysis robust, although this study is observational in nature. The main limitation of the study is that even though the data may reveal some associations, it has to be interpreted with caution due to the inherent limitations of the study design. I have some comments that would help to improve the overall quality of the manuscript:

-Can the authors comment how many facilities send data to the networks used to collect the data of the study?

-One important limitation for the study is that negative cases are only classified based on the data from the in-network facilities, but if a patient classified as “negative” had indeed a positive test reported in a different facility (or an in-house / over-the-counter test) it would be missed by the current study search strategy. The authors should state this important limitation in the discussion section of the paper.

-As the authors state, an important limitation is the fact that other infections can account for the PASC conditions. However, the authors should also mention that non-infectious conditions could account for some of these conditions, and that’s another limitation as the authors did not control for comorbid conditions.

-What does the U099/B948 code mean?

-The authors should also state in their limitations that the imputation of a missing ADI value with median ADI per site may be not an accurate estimation.

-Can the authors comment on the fact that the number of patients with positive tests during the time the Delta variant circulated was significantly lower than the number of positive tests during the time of the ancestral variant? As the Delta variant is believed to have spread faster, it is really surprising the authors found these differences. If this is related to the fact that most positive patients for the ancestral variant were diagnosed in NY state, the authors should elaborate on this. The same is true for the negative cases. There seems to be a significantly lower number of cases tested during the Delta period.

-Could the authors provide the data of how many positive cases and how many negative cases in each variant wave (ancestral vs. delta variant) were from New York, and how many were from Florida?

-I am not sure if the following sentence is appropriate for the discussion “In the absence of hypoxemia, no current treatment exists for persistent dyspnea, although novel strategies to help patients, such as breathlessness training, are being evaluated.18”it feels a little bit out of context to me.

6. PLOS authors have the option to publish the peer review history of their article (what does this mean?). If published, this will include your full peer review and any attached files.

Reviewer #1: No

Reviewer #2: No

---

## [Author Response · Author response to Decision Letter 0]

28 Nov 2023

RESPONSE TO REVIEWERS

Journal Requirements:

Response: We have revised accordingly.

Response: We have inserted the following sentence into the section on Ethical Review: “The IRBs waived informed consent for this observational cohort study that involves a limited, but not de-identified, data set.”

"This research was funded by the National Institutes of Health (NIH) Agreement OTA OT2HL161847 as part of the Researching COVID to Enhance Recovery (RECOVER) research program."

"This study is part of the NIH Researching COVID to Enhance Recovery (RECOVER) Initiative, which seeks to understand, treat, and prevent the post-acute sequelae of SARS-CoV-2 infection (PASC). This research was funded by the National Institutes of Health (NIH) Agreement OTA OT2HL161847 as part of the Researching COVID to Enhance Recovery (RECOVER) research program. NIH played a role in evaluating and developing the overall structure of the RECOVER research program, but not in the design and analysis of this specific study."

Response: We have revised accordingly.

4. One of the noted authors is a group or consortium "the RECOVER Consortium". In addition to naming the author group, please list the individual authors and affiliations within this group in the acknowledgments section of your manuscript. Please also indicate clearly a lead author for this group along with a contact email address

Response: Consortium members are now listed in the Acknowledgements. Corresponding author listed on title page.

Response: Data availability now addressed in Cover Letter.

Reviewers' comments:

RESPONSE TO REVIEWERS

Reviewer #1: Thank you for the opportunity to review this manuscript. It is generally well written and clear, so my comments focus primarily on a few methodological questions and presentation. In the methods, the authors state, “Patients were defined as having a comorbidity… 2 corresponding diagnoses documented during the baseline period.” I believe that I read earlier that the baseline period was as short as 7 days. Is it possible to have had 2 corresponding diagnoses during that short a time period? 

Response: The clinical features were collected during the baseline period, which is defined as 3 years to 1 week before lab-confirmed SARS-CoV-2 infection. We have revised the texts in Methods-Cohort Enrollment and Follow-up, and Methods-Data Analysis subsections accordingly to reflect this.

Reviewer #1: This is minor, but it does not make sense to me. Also, earlier in that paragraph “inpatients” should be “inpatient.”

Response: We have changed inpatients to inpatient in the Methods-Data Analysis subsection.

Reviewer #1: Which covariates from Table 1 were included in the hazards ratio calculations? These should be stated in the methods.

Response: We used all the covariates in Table 1 for our adjusted analyses following our previous work 17,22. We have revised the paragraph as follows: 

Both the aHR and excess burden calculations used the same covariates for adjustment. We summarized the baseline covariates in Table 1 and have included detailed descriptions in our previous studies. 17,22 Baseline covariates included age, gender, race, ethnicity, and additional factors described here. The national-level area deprivation index (ADI) was used to assess the socioeconomic disadvantage of patients. We imputed a missing ADI value with median ADI per site. Healthcare utilization was measured as the number of inpatient, outpatient, and emergency encounters (0, 1-2, 3-4, 5 or more visits for each encounter type). The Body Mass Index (BMI) was categorized according to WHO guidelines. We adopted a tailored list of the Elixhauser comorbidities and related drug categories (e.g., corticosteroid and immunosuppressant prescriptions) to capture comorbidities. Patients were defined as having comorbidity if they had at least two corresponding diagnoses documented during the baseline period. The clinical features were collected during the baseline period, namely 3 years to 1 week before lab-confirmed SARS-CoV-2 infection.

17Zang, Chengxi, Yongkang Zhang, Jie Xu, Jiang Bian, Dmitry Morozyuk, Edward J. Schenck, Dhruv Khullar et al. "Data-driven analysis to understand long COVID using electronic health records from the RECOVER initiative." Nature Communications 14, no. 1 (2023): 1948.

22 Zhang, Hao, Chengxi Zang, Zhenxing Xu, Yongkang Zhang, Jie Xu, Jiang Bian, Dmitry Morozyuk et al. "Data-driven identification of post-acute SARS-CoV-2 infection subphenotypes." Nature Medicine 29, no. 1 (2023): 226-235.”

Reviewer #1: The excess burden is quite different between the two time periods. Is it possible that more people died during the ancestral period, thus did not develop PASC? Were they included if they subsequently died? Was death considered a PASC? In any event, it seems that these differences in the magnitude of excess burden bears more discussion.

Response: We did not consider death as a component of PASC. We analyzed PASC in patients who were still alive beyond their acute infection period, namely alive beyond + 30 days, and we modeled death as a competing risk in our analyses. We further reported death data in the follow-up period (+30 – 180 days) in Table 1. We added text about this to the Discussion-Limitations paragraph.

 Entire Study Period Ancestral Strain Period Delta Period

Characteristics Cases Control Cases Control Cases Control

Death in follow-up (%) 724 (1.3) 4,676 (0.9) 275 (1.4) 2,665 (1.2) 102 (1.3) 380 (0.7)

Reviewer #1: Figure 1 Legend: “Patient” should be “Patients,” suggest adding “(index date)” after “SARS-CoV-2.” 

Response: We have revised accordingly.

Reviewer #1: The text about measurement dates following the title should be included in the methods.

Response: We have added the following description to the Methods section, “The baseline period was defined as 3 years to 1 week before lab-confirmed SARS-CoV-2 infection and the follow-up period was defined as 31 to 180 days after the index date.17,22”

Reviewer #1: Figure 1: There seem to be connector lines missing below the SARS cases and controls boxes and individual connector lines missing below the 6 boxes on the next row.

Response: We have re-designed the Fig1.

Reviewer #1: Figure 2: What is the significance of the dotted lines?

Response: We have re-designed Fig2 and used different color panels (separated by the dotted lines) to represent different organ systems, including (from top to bottom): the nervous system or mental disorders, skin, respiratory system, circulatory system, blood, endocrine and metabolic, digestive system, and other signs. 

Reviewer #1: Figure 3: This figure is pretty, but a bit confusing. I do not believe that the figure of the person is necessary and should come out. You could more clearly present the data by adding a column to the left of the “all” column that indicates the system for each group of symptoms/conditions and also remove the color from those boxes. It might be helpful to state that the figure is a heat map and indicate that the “thermometer” is the legend.

Response: We have re-designed Fig3 and revised the associated caption texts.

Reviewer #1: I did not see reference to Supplemental Fig 2 in the text. Did I miss it?

Response: Thanks for your question. We have added the reference to Supp Fig 2 in the result-section and discussion section. 

 

Reviewer #2: Can the authors comment how many facilities send data to the networks used to collect the data of the study? 

Response: We have revised our Methods text to incorporate these information: “…INSIGHT, which contains records from approximately 12 million persons who received services across five health systems (Albert Einstein School of Medicine/Montefiore Medical Center, Columbia University and Weill Cornell Medicine/New York-Presbyterian Hospital, lcahn School of Medicine/Mount Sinai Health System, and New York University School of Medicine/Langone Medical Center) in the New York City (NYC) metropolitan area, and OneFlorida+, which contains records from approximately 15 million persons receiving services across 13 health systems (University of Florida and UF Health, Florida State University, University of Miami and UHealth, Orlando Health System, AdventHealth, Tallahassee Memorial HealthCare, Tampa General Hospital, Bond Community Health Center Inc., Nicklaus Children’s Hospital, CommunityHealth IT, University of South Florida and USF Health, University of Alabama at Birmingham, Emory University) in Florida.”

Reviewer #2: One important limitation for the study is that negative cases are only classified based on the data from the in-network facilities, but if a patient classified as “negative” had indeed a positive test reported in a different facility (or an in-house / over-the-counter test) it would be missed by the current study search strategy. The authors should state this important limitation in the discussion section of the paper.

Response: We have added this limitation in the discussion section: “…Our study is subject to important limitations. First, we may have misclassified patients as not infected with COVID-19 because a test was never performed or not recorded in the in-network facilities. This may have been more likely during the follow-up of the post-Delta wave, as the follow-up period overlapped with the increasing availability of home testing. Such misclassification would likely lead us to underestimate the prevalence of PASC or underestimate the relative risk of PASC, particularly during the first wave when diagnostic testing was less widely available. …” 

Reviewer #2: As the authors state, an important limitation is the fact that other infections can account for the PASC conditions. However, the authors should also mention that non-infectious conditions could account for some of these conditions, and that’s another limitation as the authors did not control for comorbid conditions.

Response: We are not sure which conditions the reviewer is referring to, because we compared the new onset of non-infectious conditions (and symptoms) between those with a COVID-19 diagnosis and those without during the study period. If other non-infectious conditions also mimic PASC, we would expect this to bias our results toward no or less association between COVID-19 and PASC.

Reviewer #2: What does the U099/B948 code mean?

Response: We used ICD-10 diagnostic codes B948 (sequelae of other specified infectious and parasitic diseases) and U099 (post-COVID-19 condition, unspecified) to capture general or unspecified PASC conditions. We added this to the Methods-Data Analysis section.

Reviewer #2: The authors should also state in their limitations that the imputation of a missing ADI value with median ADI per site may be not an accurate estimation.

Response: Thanks for your suggestions. We have added this to the Discussion-Limitations paragraph: “…While we adjusted for many characteristics, other unmeasured factors or missing values (e.g., missing ADI value, smoking status, etc.) could explain differences…”

Reviewer #2: Can the authors comment on the fact that the number of patients with positive tests during the time the Delta variant circulated was significantly lower than the number of positive tests during the time of the ancestral variant? As the Delta variant is believed to have spread faster, it is really surprising the authors found these differences. If this is related to the fact that most positive patients for the ancestral variant were diagnosed in NY state, the authors should elaborate on this. The same is true for the negative cases. There seems to be a significantly lower number of cases tested during the Delta period.

Response: We covered site-specific analyses in Supplementary materials (Fig S1 for temporal dynamics, Fig S2 for the excess burdens, and Table S1 for the number of patients in different period). We reference this issue in the Discussion section, “Third, the number of cases during the Delta variant period was substantially higher in Florida (Supplementary Figure S1 and Supplementary Table S1), and, for several conditions, the magnitude of excess burden varied between NY and Florida (Supplementary Figure S2).” In our sensitivity analyses, we observed magnitude differences but the major conclusions still held.

Reviewer #2: Could the authors provide the data of how many positive cases and how many negative cases in each variant wave (ancestral vs. delta variant) were from New York, and how many were from Florida?

Response: We have incorporated this information in the supplemental data, see below.

Table. S1. Number of patients in All time (March 2020 to November 2021) versus Ancestral Strain Period (March 2020 to September 2020) versus Delta Variant Period (June 2021 to November 2021), stratified by different regions, NYC Insight vs. Florida OneFlorida.

 Insight OneFlorida

 All SARS-CoV-2 Positive SARS-CoV-2 Negative All SARS-CoV-2 Positive SARS-CoV-2 Negative

All time 361,401 35,275 326,126 199,351 22,341 177,010

Ancestral Strain Period 149,734 12,611 137,123 86,051 7,332 78,719

Delta Period 30,491 2,035 28,456 30,323 6,062 24,261

Reviewer #2: I am not sure if the following sentence is appr

---

## [Decision Letter · Decision Letter 1]

11 Jan 2024

PONE-D-23-04388R1Excess burden of respiratory and abdominal conditions following COVID-19 infections during the ancestral and Delta variant periods in the United States: An EHR-based cohort study from the RECOVER ProgramPLOS ONE

Dear Dr. Varma,

Thank you for submitting your manuscript to PLOS ONE. After careful consideration, we feel that it has merit but does not fully meet PLOS ONE’s publication criteria as it currently stands. Therefore, we invite you to submit a revised version of the manuscript that addresses the points raised during the review process.

The requested changes have been made. However, some minor fixes are missing. Please follow what the reviewers suggest.

We look forward to receiving your revised manuscript.

Kind regards,

Roberto Scendoni

Academic Editor

PLOS ONE

Journal Requirements:

Reviewers' comments:

Reviewer's Responses to Questions

**Comments to the Author**

1. If the authors have adequately addressed your comments raised in a previous round of review and you feel that this manuscript is now acceptable for publication, you may indicate that here to bypass the “Comments to the Author” section, enter your conflict of interest statement in the “Confidential to Editor” section, and submit your "Accept" recommendation.

Reviewer #1: (No Response)

Reviewer #3: (No Response)

2. Is the manuscript technically sound, and do the data support the conclusions?

Reviewer #1: Yes

Reviewer #3: (No Response)

3. Has the statistical analysis been performed appropriately and rigorously? 

Reviewer #1: Yes

Reviewer #3: (No Response)

4. Have the authors made all data underlying the findings in their manuscript fully available?

Reviewer #1: Yes

Reviewer #3: (No Response)

5. Is the manuscript presented in an intelligible fashion and written in standard English?

Reviewer #1: Yes

Reviewer #3: (No Response)

6. Review Comments to the Author

Reviewer #1: The authors have addressed my concerns, but on Figure 1, the last two boxes indicate that they were patients without targeted diagnosis. Just confirming that this should not be with targeted diagnosis.

Reviewer #3: Indeed, it is acceptable to request only a minor revision, specifying the need to expand the bibliography, as it is currently too concise. Specifically, include reviews on coinfections in hospitalized individuals, along with a brief comment (e.g., doi: 10.3389/fmed.2021.681469; doi: 10.3390/pathogens12050646; doi: 10.1186/s13054-023-04312-0; doi: 10.1016/j.ijid.2020.10.040).

7. PLOS authors have the option to publish the peer review history of their article (what does this mean?). If published, this will include your full peer review and any attached files.

Reviewer #1: No

Reviewer #3: No

---

## [Author Response · Author response to Decision Letter 1]

14 Jan 2024

Reviewer #1: The authors have addressed my concerns, but on Figure 1, the last two boxes indicate that they were patients without targeted diagnosis. Just confirming that this should not be with targeted diagnosis. 

Response: The Figure is labeled correctly. The population being studied is patients without one of the PASC diagnoses at baseline, and, within that population, we are comparing those with at COVID-19 diagnosis to those without a COVID-19 diagnosis.

Reviewer #3: Indeed, it is acceptable to request only a minor revision, specifying the need to expand the bibliography, as it is currently too concise. Specifically, include reviews on coinfections in hospitalized individuals, along with a brief comment (e.g., doi: 10.3389/fmed.2021.681469; doi: 10.3390/pathogens12050646; doi: 10.1186/s13054-023-04312-0; doi: 10.1016/j.ijid.2020.10.040). 

Response: We have added the requested references.

---

## [Editor Report · Decision Letter 2]

16 Jan 2024

Excess burden of respiratory and abdominal conditions following COVID-19 infections during the ancestral and Delta variant periods in the United States: An EHR-based cohort study from the RECOVER Program

PONE-D-23-04388R2

Dear Dr. Varma,

We’re pleased to inform you that your manuscript has been judged scientifically suitable for publication and will be formally accepted for publication once it meets all outstanding technical requirements.

Kind regards,

Roberto Scendoni

Academic Editor

PLOS ONE
---

## [Editor Report · Acceptance letter]

9 May 2024

PONE-D-23-04388R2 

PLOS ONE

Dear Dr. Varma, 

I'm pleased to inform you that your manuscript has been deemed suitable for publication in PLOS ONE. Congratulations! Your manuscript is now being handed over to our production team.

Kind regards, 

on behalf of

Dr. Roberto Scendoni 

Academic Editor

PLOS ONE